# Data-efficient Supervised Learning is Powerful for Neural Combinatorial Optimization

## Abstract

Neural combinatorial optimization (NCO) is a promising learning-based approach to solve difficult combinatorial optimization problems. However, how to efficiently train a powerful NCO solver remains challenging. The widely-used reinforcement learning method suffers from sparse rewards and low data efficiency, while the supervised learning approach requires a large number of high-quality solutions. In this work, we develop efficient methods to extract sufficient supervised information from limited labeled data, which can significantly overcome the main shortcoming of supervised learning. For traveling salesman problem (TSP), a representative combinatorial optimization problem, we propose a set of efficient data augmentation methods and a novel bidirectional loss to better leverage the equivalent properties of problem instances, which finally lead to a promising supervised learning approach. The thorough experimental studies demonstrate our proposed method can achieve state-of-the-art performance on TSP only with a small set of 50, 000 labeled instances, while it also achieves promising generalization performances on tasks with different sizes or different distributions. We believe this somewhat surprising finding could lead to valuable rethinking on the value of efficient supervised learning for NCO.

## 1 Introduction

Many real-world applications involve challenging combinatorial optimization problems, which could be NP-hard and cannot be exactly solved in a reasonable time (Papadimitriou & Steiglitz, 1998). The traditional approach needs to design handcrafted heuristic rules for each specific problem, and requires a long search process to solve every problem instance even when they are similar to each other (Korte et al., 2011). In recent years, many learning-based algorithms have been proposed to efficiently find a good approximate solution for a given problem instance (Bengio et al., 2021). In this work, we focus on the neural combinatorial optimization (NCO) approach (Bello et al., 2016) since it can directly generate an approximate solution in real-time without any expert knowledge or predefined heuristic rules.

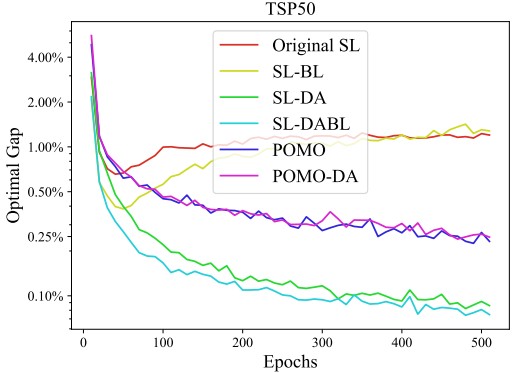

Figure 1: The optimality gap of models trained with different training strategies on the validation set.

Although a combinatorial optimization problem could be NP-hard, a real-world application could typically only care about a small subset of instances (Bengio et al., 2021). Therefore, it is possible to leverage the similar patterns shared by these instances to learn an efficient neural combinatorial solver (Vinyals et al., 2015). Supervised learning (SL) and reinforcement learning (RL) are the two main methods for training the NCO solver, which learn the pattern directly from high-quality solutions (Vinyals et al., 2015) or through extensive interaction with the environment (e.g., the problem instances) (Bello et al., 2016).

It is challenging to efficiently train a powerful NCO solver. The RL method suffers from the issues of sparse rewards (Vecerik et al., 2017; Hare, 2019) and low data efficiency (Laskin et al., 2020), which could require a huge computational budget and lead to extremely long training time (e.g., more than a week) (Joshi et al., 2020; Kwon et al., 2020). By directly learning from high-quality solutions at each step, the SL method has better sample efficiency and is a promising alternative for training an NCO solver (Joshi et al., 2019; 2020). Nevertheless, SL suffers from the difficulty of collecting sufficient labeled data (i.e., optimal or near-optimal solutions of combinatorial optimization instances). In addition, there are also some concerns on the generalization performance of the NCO solver trained by the SL method (Joshi et al., 2020).

In this work, we investigate how to overcome the shortcomings of SL-based NCO training. By leveraging the equivariance and symmetries of the problem instances and solutions, we develop novel approaches to extract sufficient information from limited high-quality solutions for data-efficient supervised learning, and we demonstrate that training POMO (Kwon et al., 2020) through our method is better than reinforcement learning. Our main contributions can be summarized as follows:

- We design four simple yet efficient data augmentation approaches to significantly enlarge the training set from limited high-quality solutions, and develop a novel bidirectional supervised loss to leverage the equivalence of solutions to further improve the training efficiency for supervised learning. With these two powerful methods, we propose a novel Supervised Learning with Data Augmentation and Bidirectional Loss (SL-DABL) algorithm for TSP.

- We conduct thorough experiments to study the efficiency of our proposed method. The results confirm that SL-DABL can achieve ***state-of-the-art performance*** on TSP with only $50,000$ training instances, and also has ***promising generalization performance*** to real-world instances with different sizes. These findings lead us to rethink some current beliefs on NCO for TSP, such as (Joshi et al., 2020).

Our findings reported in this work could be somewhat surprising and opposite to some current beliefs about the NCO method. We show that 1) the huge supervised data requirement (a major drawback) is indeed not necessary for SL and 2) RL is not always the best choice for training a NCO model. We hope they could be helpful for rethinking the role and value of efficient SL-based NCO training.

## 2 RELATED WORKS

In the past few years, many promising learning-based approaches have been proposed to tackle different combinatorial optimization problems. We briefly review the neural combinatorial optimization methods that are closely related to this work, and refer readers to Bengio et al. (2021) and Cappart et al. (2021) for comprehensive surveys.

### 2.1 SUPERVISED LEARNING FOR NCO

Vinyals et al. (2015) proposed the Pointer Network with RNN encoder-decoder structure and attention mechanism to solve TSP in an autoregressive manner. Milan et al. (2017) found that it is costly to generate enough high-quality solutions to serve as a training dataset for supervised learning, and proposed to update the initial dataset with superior solutions generated during the training process. Joshi et al. (2019) trained a graph neural network to predict the heatmap for each instance with non-autoregressive decoding. The heatmap measures the probability that each edge will belong to the optimal solution, which can be converted to a valid solution with beam search, Monte Carlo tree search (Fu et al., 2021), guided local search (Hudson et al., 2021), and dynamic programming (Kool et al., 2021).

Joshi et al. (2020) systematically studied the performance of different learning methods on both autoregressive and non-autoregressive models. This work focuses on the construction-based autoregressive model. According to the results in (Joshi et al., 2020), even with $1,280,000$ training instances, the SL approach will still be outperformed by the RL approach on the zero-shot greedy prediction for both testing and generalization performance. In this work, we propose a novel data-efficient SL method to achieve state-of-the-art performance with $50,000$ training instances, which is only $4\%$ of the training dataset in Joshi et al. (2020).

## 2.2 REINFORCEMENT LEARNING FOR NCO

Many RL-based methods have been proposed to train NCO solvers (Bello et al., 2016; Khalil et al., 2017; Nazari et al., 2018; Deudon et al., 2018; Ma et al., 2019). Kool et al. (2018) proposed the Attention Model (AM) framework to solve different vehicle routing problems. Different follow-up works have been developed to improve the AM performance with diverse solution generations (Xin et al., 2021; Kim et al., 2021). Kwon et al. (2020) proposed the POMO method with multiple greedy rollouts to leverage the multiple optima property, which is the current state-of-the-art RL algorithm for NCO. In this work, with the same model and inference strategy, our proposed SL-DABL method can outperform the RL counterpart in POMO for both testing and generalization performance.

Data augmentation is a widely-used approach to increase the amount of training data for supervised learning (Shorten & Khoshgoftaar, 2019; Feng et al., 2021). Kwon et al. (2020) have discussed how to use the data augmentation methods to improve the inference performance for NCO but not for SL-based training. Recently, Geisler et al. (2022) have shown that augmenting adversarial examples into training could improve the NCO solver's robustness with small perturbations. In this work, *we show that data augmentation is crucial for data-efficient SL-based NCO, but has little effect on RL-based training.* A concurrent work (Kim et al., 2022) used contrastive learning to leverage the symmetric property of CO problems and solutions for better instance representation, which can improve the RL-based training. We believe these two methods are indeed orthogonal and could be complementary to each other.

Equivariant neural networks (Thomas et al., 2018; Satorras et al., 2021), which can directly incorporate the equivariance properties into the model structure, is a strong alternative to data augmentation. In this work, instead of building more powerful models, we focus on improving the supervised learning approach, and study whether it can outperform the reinforcement learning counterpart on the same model. Therefore, we left the study of equivariant neural networks for NCO to future work.

## 3 DATA AUGMENTATION

In this section, we propose four simple yet efficient data augmentation (DA) approaches to significantly enlarge the training set from limited high-quality solutions. By leveraging the translation invariance property of the problem instances, our proposed SL-DA method can extract sufficient information from the small dataset to train a powerful NCO solver as shown in Figure 2.

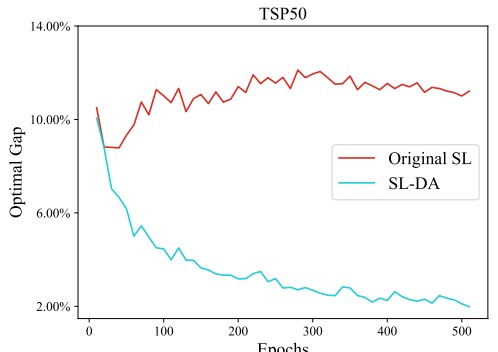

### 3.1 PRELIMINARIES: TSP

TSP is the basic case of all the vehicle routing problems, which aims to find the shortest trajectory $\tau$ from a fully connected graph $\mathbf{S}$. In this work, we focus on the 2D Euclidean TSP, as it is a common benchmark for most NCO algorithms. The fully connected graph $\mathbf{S} = (\mathbf{s}_1, \mathbf{s}_2, \ldots, \mathbf{s}_n)$ can be represented by a set of $n$ nodes in normalized 2D Cartesian coordinate, i.e., $\mathbf{s}_i = (x_i, y_i)^\intercal$ with $x_i \in [0, 1]$ and $y_i \in [0, 1]$ for $i = \{1, 2, \ldots, n\}$.

Figure 2: Supervised learning with/without efficient data augmentation on TSP50. Both methods only have $50,000$ training instances. The vanilla SL method struggles to train the NCO solver, while our proposed SL-DA method has much better performance. The experimental setting for this analysis can be found in Section 3.6.

The outputted trajectory $\tau = (t_1, t_2, \ldots, t_n)$ is a permutation of node indices, i.e., $t_i, t_j \in \{1, 2, \ldots, n\}$ and $t_i \neq t_j$ for any two different indices $i, j \in \{1, 2, \ldots, n\}$. The cost of $\tau$ is calculated as

$$c(\tau|\mathbf{S}) = \sum_{i=1}^{n-1} \|\mathbf{s}_{t_{i+1}} - \mathbf{s}_{t_i}\|_2 + \|\mathbf{s}_{t_n} - \mathbf{s}_{t_1}\|_2.$$

In this work, we represent the graph $\mathbf{S}$ as $[\mathbf{x}; \mathbf{y}]$, where $\mathbf{x} = [x_1, x_2, \ldots, x_n]$ and $\mathbf{y} = [y_1, y_2, \ldots, y_n]$.

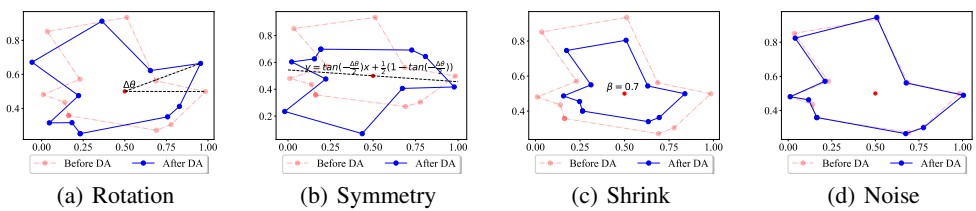

(a) Rotation  (b) Symmetry  (c) Shrink  (d) Noise

Figure 3: Examples of generating a new instance via the **(a)** rotation operator, **(b)** symmetry operator, **(c)** the shrink operator, and **(d)** the noise operator, respectively.

## 3.2 ROTATION

Each TSP instance is a fully connected graph whose optimal solution is invariant with the rotation of the entire graph. Therefore, with a single optimal solution to a given instance, we can generate multiple new instances by randomly rotating the original instance at different angles. The details of this rotation operator are provided in Algorithm 1.

---

**Algorithm 1** Rotation

**Input:** the original graph $\mathbf{S}$.
**Output:** the augmented graph $\mathbf{S}'$.

1: $[\mathbf{x}_m; \mathbf{y}_m] \leftarrow [\mathbf{x} - 0.5; \mathbf{y} - 0.5]$;
2: $\boldsymbol{\rho}, \boldsymbol{\theta} \leftarrow$ Cartesian2Polar$(\mathbf{x}_m, \mathbf{y}_m)$;
3: $\Delta\theta \sim \mathcal{U}(0, 2\pi)$;
4: $\boldsymbol{\theta}' \leftarrow \boldsymbol{\theta} + \Delta\theta$;
5: $[\mathbf{x}_m; \mathbf{y}_m] \leftarrow$ Polar2Cartesian$(\boldsymbol{\rho}, \boldsymbol{\theta}')$;
6: $\mathbf{S}' \leftarrow [\mathbf{x}_m + 0.5; \mathbf{y}_m + 0.5]$;

---

**Algorithm 2** Symmetry

**Input:** the original graph $\mathbf{S}$.
**Output:** the augmented graph $\mathbf{S}'$.

1: $[\mathbf{x}_m; \mathbf{y}_m] \leftarrow [\mathbf{x} - 0.5; \mathbf{y} - 0.5]$;
2: $\boldsymbol{\rho}, \boldsymbol{\theta} \leftarrow$ Cartesian2Polar$(\mathbf{x}_m, \mathbf{y}_m)$;
3: $\Delta\theta \sim \mathcal{U}(0, 2\pi)$;
4: $\boldsymbol{\theta}' \leftarrow -(\boldsymbol{\theta} + \Delta\theta)$;
5: $[\mathbf{x}_m; \mathbf{y}_m] \leftarrow$ Polar2Cartesian$(\boldsymbol{\rho}, \boldsymbol{\theta}')$;
6: $\mathbf{S}' \leftarrow [\mathbf{x}_m + 0.5; \mathbf{y}_m + 0.5]$;

---

As shown in lines 1-2 of Algorithm 1, we first move the graph by changing its center from $(0.5, 0.5)^\intercal$ to $(0, 0)^\intercal$ and express the nodes in terms of the polar coordinates as

$$\boldsymbol{\rho} = \sqrt{\mathbf{x}_m^2 + \mathbf{y}_m^2}, \quad \boldsymbol{\theta} = \arctan \frac{\mathbf{y}_m}{\mathbf{x}_m}.$$

Then, we randomly generate an angle $\Delta\theta \in (0, 2\pi)$ then add it to the current $\boldsymbol{\theta}$ to get a new instance (i.e., lines 3-4 of Algorithm 1). After that, we transform the new instance into the Cartesian coordinate system as

$$\mathbf{x}_m = \boldsymbol{\rho}\cos\boldsymbol{\theta}', \quad \mathbf{y}_m = \boldsymbol{\rho}\sin\boldsymbol{\theta}'.$$

Finally, we move the graph back to the location centered at $(0.5, 0.5)^\intercal$ and output the augmented graph $\mathbf{S}'$.

Figure 3(a) illustrates an example of generating a new instance via the rotation operator. It is clear that the new instance has the same optimal solution with the original one.

## 3.3 SYMMETRY

Similar to rotation, the optimal solution is also invariant to the symmetry operator. Since all nodes are located in $[0, 1]^2$, for uniformity, we can set the axis of symmetry as any line that passes through the midpoint $(0.5, 0.5)^\intercal$.

Algorithm 2 presents the symmetry operator in detail. Compared with the rotation operator, it additionally flips the rotated graph along the horizontal axis of the polar coordinate system on line 4. The symmetry axis between the new graph and the original graph can be expressed as

$$y = kx + \frac{1}{2}(1 - k), \quad \text{where } k = \tan\left(-\frac{\Delta\theta}{2}\right).$$

An example of generating the new instance via the symmetry operator is depicted in Figure 3(b). We can see that both the original instance and the new instance have the same optimal solution.

### 3.4 SHRINK

| **Algorithm 3** Shrink | **Algorithm 4** Noise |
|---|---|
| **Input:** original graph $\mathbf{S}$, threshold parameter $\gamma$. | **Input:** original graph $\mathbf{S}$. |
| **Output:** augmented graph $\mathbf{S}'$ | **Output:** augmented graph $\mathbf{S}'$. |
| 1: $[\mathbf{x}_m; \mathbf{y}_m] \leftarrow [\mathbf{x} - 0.5; \mathbf{y} - 0.5]$; | 1: $d \leftarrow \text{NodesMinimumDistance}(\mathbf{x}, \mathbf{y})$; |
| 2: $\boldsymbol{\rho}, \boldsymbol{\theta} \leftarrow \text{Cartesian2Polar}(\mathbf{x}_m, \mathbf{y}_m)$; | 2: $\mathbf{r}_\rho \sim \mathcal{U}(0, 1)$; |
| 3: $\beta \sim \mathcal{U}(1 - \gamma, 1 + \gamma)$; | 3: $\mathbf{r}_\theta \sim \mathcal{U}(0, 2\pi)$; |
| 4: $\boldsymbol{\rho}' \leftarrow \beta \boldsymbol{\rho}$; | 4: $\mathbf{r}'_\rho \leftarrow \frac{d}{2} \mathbf{r}_\rho$; |
| 5: $[\mathbf{x}_m; \mathbf{y}_m] \leftarrow \text{Polar2Cartesian}(\boldsymbol{\rho}', \boldsymbol{\theta})$; | 5: $\mathbf{r}_x, \mathbf{r}_y \leftarrow \text{Polar2Cartesian}(\mathbf{r}'_\rho, \mathbf{r}_\theta)$; |
| 6: $\mathbf{S}' \leftarrow [\mathbf{x}_m + 0.5; \mathbf{y}_m + 0.5]$; | 6: $\mathbf{S}' \leftarrow [\mathbf{x} + \mathbf{r}_x; \mathbf{y} + \mathbf{r}_y]$; |

The shrink operator linearly scales the original graph. Since the relative positions of all nodes are not changed, the optimal solution of the new instance is the same as the original one. Similar to the rotation and symmetry operators, we use the midpoint $(0.5, 0.5)^\intercal$ as the center for the shrink operator. The detailed procedure is given in Algorithm 3.

We first move the given instance to the new location such that its midpoint is $(0, 0)^\intercal$ as in line 1, and map all nodes into the polar coordinate system as shown in line 2 of Algorithm 3. Then a coefficient $\beta$ is randomly sampled from $\mathcal{U}(1 - \gamma, 1 + \gamma)$ to control the zoom degree. The predefined threshold parameter $\gamma$ can prevent the graph from zooming to an extremely small or large scale. In this paper, we set $\gamma$ to 0.3. The final step of Algorithm 3 restores the scaled graph to the original Cartesian coordinate.

In Figure 3(c), an example of generating a new instance via the shrink operator is provided, where both the original and shrinking instance share the same optimal solution.

### 3.5 NOISE

Unlike the above three operators that perfectly preserve the relative positional relationship, the noise operator generates new graphs by randomly perturbing each node in the original graph. In other words, without any further restriction, the newly generated instances could have different optimal solutions to the original instance. This property is undesirable for data augmentation.

However, the optimal solution could still keep the same if we only slightly change the node coordinates while *qualitatively* maintaining the relative position. To be specific, we can add a small enough noise to each node to perturb the graph such that the noise's upper bound is half the minimum distance between each pair of nodes $\frac{d}{2}$. In this way, the nodes after perturbation will be in a small region around their original location and not overlap with each other. This tight restriction can also guarantee the newly generated instance will still have the same optimal solution as the original instance.

Algorithm 4 describes the noise operator in detail. As shown in line 1, the minimal distance between each pair of nodes is calculated first. In lines 2-3, two coefficient vectors $\mathbf{r}_\rho \in (0, 1)^n$ and $\mathbf{r}_\theta \in (0, 2\pi)^n$ are randomly sampled to determine the magnitude and direction of the noise, respectively. Finally, we add the corresponding movements over the x and y axes as the noise for each node. This noise operator can adaptively set the noise upper bound and generate new graphs without changing the optimal solution. As the example illustrated in Figure 3(d), the optimality of the original label solution is guaranteed.

### 3.6 COMPARISON

These four DA operators can be applied independently or stacked together in a specific manner to generate new problem instances to support SL-based training. They are also model-agnostic and can be used to train any SL-based NCO solver. In this subsection, we investigate their effectiveness for training the AM solver with the implementation in Kwon et al. (2020).

We compare the performance of the models trained by SL with different DA operators on the validation set, which consists of 1,000 randomly generated instances. Both training and validation are

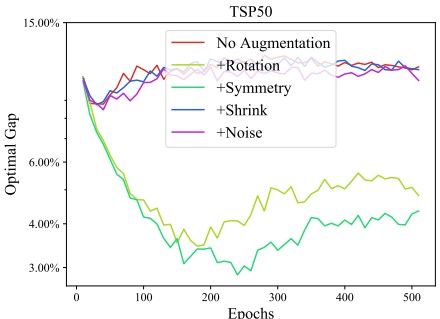
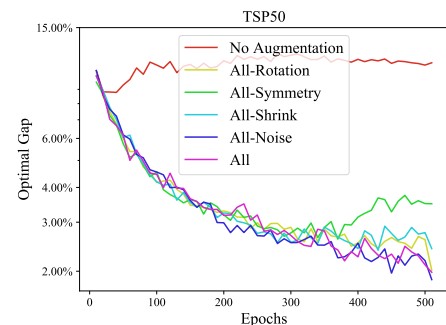

Figure 4: The average optimal gap on the validation set generated by the Attention Model trained with supervised learning using each individual DA operator.

Figure 5: The average optimal gap on the validation set generated by the Attention Model trained with supervised learning using combined DA operators.

50-node TSP instances (denoted as TSP50). The performance is evaluated by the average optimal gap, and the optimal values are calculated by Concorde (Applegate et al., 2006).

As shown in Figure 4, each of the proposed DA operators can improve the performance compared to using the original training dataset directly. Specifically, the rotation and symmetry operators can remarkably alleviate the overfitting problem as well as reduce the optimal gap. In contrast, the effectiveness of using the shrink operator or the noise operator separately is not significant.

In addition to the performance assessment of each individual DA operator, we also investigate the efficiency of the combined DA operators. We stack all DA operators first and remove one of them for enumeration. For instance, the *All-Rotation* represents we employ all DA operators except the rotation operator. Since there is overlap in rotation and symmetry operators, we randomly employ one of them when using them both.

As shown in Figure 5, all kinds of combinations can significantly reduce the optimal gap. The symmetry operator appears to be the most effective one, but the other operators also contribute to the performance improvement to varying degrees. In this paper, we adopt the combination of all four DA operations as our DA strategy.

## 4 BIDIRECTIONAL LOSS

In this section, by leveraging the equivalence of optimal solutions, we propose a novel bidirectional loss to further improve the data efficiency for SL-based training.

### 4.1 LOSS FUNCTIONS

The construction-based NCO solver sequentially generates the solution in an autoregressive manner, which selects one node at each step. The node selection can be viewed as a classification problem, and the goal of the SL-based method is to minimize the conditional cross-entropy loss from the optimal solution (Vinyals et al., 2015):

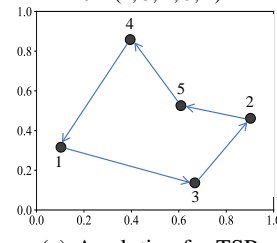

(a) A solution for TSP    (b) Equivalent optimal solutions

Figure 6: Equivalent optimal solutions for a TSP instance.

$$\mathcal{L}(\mathbf{S}, \overrightarrow{\tau}) = -\sum_{t=1}^{n} \log p_{\phi}(\overrightarrow{\tau_t}|\mathbf{S}_{\overrightarrow{\tau}_{0:t-1}})$$

where $\overrightarrow{\tau}$ is the optimal solution for the corresponding instance $\mathbf{S}$, $\mathbf{S}_{\overrightarrow{\tau}_{0:t-1}}$ is the partial tour up to step $t-1$, $\overrightarrow{\tau}_t$ is the selected node at step $t$, and $\phi$ is the parameters of the training model.

However, a CO problem instance could have multiple optimal solutions (Kwon et al., 2020; Kotary et al., 2021). As shown in Figure 6, for a TSP instance with $n$ nodes and a given optimal solution $\overrightarrow{\tau}$,

we can have $n$ equivalent optimal solutions $\overrightarrow{\tau}^i$ with different starting node $i$. In addition, since the solution (tour) can move along the reverse direction, we can also have the other $n$ optimal solution $\overleftarrow{\tau}^i$. Therefore, there are total $2n$ equivalent solutions to the single given optimal solution $\overrightarrow{\tau}$.

In this work, we propose a novel bidirectional loss function to leverage all the equivalent optimal solutions for data-efficient SL-based training:

$$\mathcal{L}_B(\mathbf{S}, \overrightarrow{\tau}) = \frac{1}{n} \sum_{i=1}^{n} A(\mathcal{L}(\mathbf{S}, \overrightarrow{\tau}^i), \mathcal{L}(\mathbf{S}, \overleftarrow{\tau}^i)),$$

where $A(\cdot, \cdot)$ denotes the aggregation function over the two reverse directions from the same starting node, which can be one of $\{\texttt{min}, \texttt{mean}, \texttt{max}\}$. The $\texttt{min}$ aggregation greedily optimizes the solution for the prefer direction, the $\texttt{mean}$ aggregation considers solutions for both directions, an the $\texttt{max}$ aggregation optimizes the upper bound (e.g., the solutions with worse performance).

## 4.2 COMPARISON

In this subsection, we compare the performance of different aggregations for the bidirectional loss with data augmentation. As shown in Figure 7, all aggregated bidirectional loss functions can outperform the original supervised learning function with the same amount of provided optimal solutions (e.g., $50,000$). Among the three different aggregation functions, the $\texttt{min}$ aggregation achieves the best performance, and we use it as the default setting in the rest of this work. An ablation study on the three aggregations with different settings can be found in Appendix.

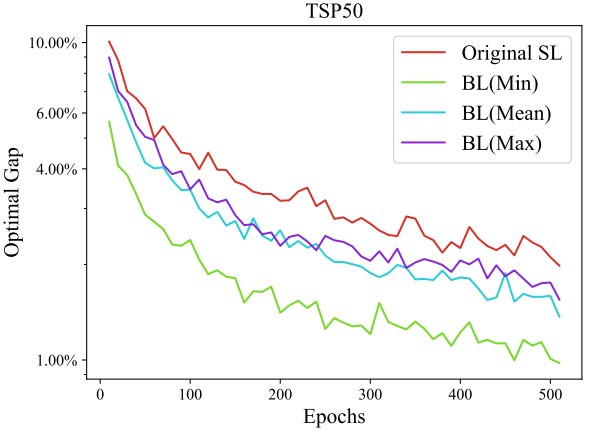

Figure 7: The average optimal gaps on the validation set with different loss functions.

## 4.3 SL-DABL

We combine the aforementioned data augmentation approaches and the bidirectional loss together to propose our Supervised Learning with Data Augmentation and Bidirectional Loss (SL-DABL) method as shown in Algorithm 5. SL-DABL is model-agnostic and can be used to train different construction-based NCO solvers. In this work, we adopt the same model structure, hyperparameter settings, as well as the inference strategy with multiple starting nodes from POMO (Kwon et al., 2020). In other words, the only difference is our proposed SL-DABL v.s. the RL-based training method with multiple rollouts developed in Kwon et al. (2020). Following POMO, we also propose a variant (denoted as SL-DABL ×8) that conducts additional ×8 instance augmentation in the inference phase.

---

**Algorithm 5** SL-DABL

1: **Input:** the training dataset $\mathcal{D}$, the number of training steps $\texttt{iter}_{\max}$, the batch size $B$, and the shrink threshold parameter $\gamma$.
2: **Output:** the trained model with parameters $\phi^*$.
3: Initialize the model with parameters $\phi$;
4: **for** $\texttt{iter} = 1, \ldots, \texttt{iter}_{\max}$ **do**
5:    $\mathbf{S}^i, \overrightarrow{\tau}^i \sim \text{SampleInstance}(\mathcal{D}) \quad \forall i \in \{1, \ldots, B\}$;
6:    $a \sim \mathcal{U}(0, 1)$;
7:    **if** $a < 0.5$ **then**
8:       $\mathbf{S}^i \leftarrow \text{Rotation}(\mathbf{S}^i)$;
9:    **else**
10:      $\mathbf{S}^i \leftarrow \text{Symmetry}(\mathbf{S}^i)$;
11:   **end if**
12:   $\mathbf{S}^i \leftarrow \text{Shrink}(\mathbf{S}^i, \gamma)$;
13:   $\mathbf{S}^i \leftarrow \text{Noise}(\mathbf{S}^i)$;
14:   $\nabla \mathcal{L}(\boldsymbol{\theta}) \leftarrow \frac{1}{B} \sum_{i=1}^{B} \nabla \mathcal{L}_B(\mathbf{S}^i, \overrightarrow{\tau}^i)$;
15:   $\phi \leftarrow \text{ADAM}(\phi, \nabla \mathcal{L}(\phi))$;
16: **end for**

---

# 5 EXPERIMENT

In this section, we first compare our SL-DABL method with other widely-used learning/non-learning solvers on the uniform TSP instances. Then we conduct ablation studies to analyze each component of SL-DABL. Finally, SL-DABL is compared against RL on the generalization ability to different problem sizes and real-world TSPLib instances.

## 5.1 OVERALL COMPARISON

| Method | Type | TSP20 | | | TSP50 | | | TSP100 | | |
|---|---|---|---|---|---|---|---|---|---|---|
| | | Cost | Gap | Time | Cost | Gap | Time | Cost | Gap | Time |
| Concorde | Solver | 3.82473 | 0.000% | 1.13m | 5.69311 | 0.000% | 6.66m | 7.76541 | 0.000% | 30.85m |
| Gurobi[†] | Solver | 3.8302 | 0.000% | 2.33m | 5.6905 | 0.000% | 26.20m | 7.7609 | 0.000% | 3.57h |
| LKH3[†] | Heuristic | 3.8303 | 0.000% | 20.96m | 5.6905 | 0.001% | 26.25m | 7.7611 | 0.003% | 49.96m |
| Wu {5000} | RL, I | 3.82551 | 0.020% | 1.07h | 5.70456 | 0.201% | 1.42h | 7.88117 | 1.491% | 1.83h |
| Costa {2000} | RL, I | 3.82479 | 0.002% | 28.45m | 5.69957 | 0.114% | 44.23m | 7.82389 | 0.753% | 1.06h |
| AM(Greedy) | RL | 3.83565 | 0.285% | 0.35m | 5.78875 | 1.680% | 0.80m | 8.10442 | 4.366% | 1.80m |
| AM(Sampling) | RL | 3.83694 | 0.319% | 0.42m | 5.80361 | 1.941% | 1.06m | 8.15186 | 4.977% | 2.05m |
| POMO | RL | 3.82557 | 0.022% | 0.01m | 5.70238 | 0.163% | 0.03m | 7.82817 | 0.808% | 0.17m |
| POMO × 8 | RL | 3.82479 | 0.002% | 0.08m | 5.69507 | 0.035% | 0.23m | 7.78218 | 0.216% | 1.08m |
| POMO-DA | RL | 3.82572 | 0.026% | 0.01m | 5.70270 | 0.168% | 0.03m | 7.84934 | 1.081% | 0.17m |
| POMO-DA × 8 | RL | 3.82480 | 0.002% | 0.08m | 5.69521 | 0.037% | 0.23m | 7.78944 | 0.309% | 1.08m |
| GCN[†] | SL | 3.86 | 0.600% | 0.10m | 5.87 | 3.100% | 0.92m | 8.41 | 8.380% | 6m |
| GCN[†] | SL, BS | 3.84 | 0.100% | 0.33m | 5.71 | 0.260% | 2m | 7.92 | 2.110% | 10m |
| GCN[†] | SL, BS* | 3.84 | 0.010% | 12m | 5.70 | 0.010% | 18m | 7.87 | 1.390% | 40m |
| SL-DABL | SL | 3.82496 | 0.006% | 0.01m | 5.69562 | 0.044% | 0.03m | 7.78793 | 0.290% | 0.17m |
| SL-DABL × 8 | SL | 3.82473 | 0.000% | 0.08m | 5.69318 | 0.001% | 0.23m | 7.76925 | 0.049% | 1.08m |

Table 1: Experiment results on various TSP instances. In the *Type* column, **RL**: Reinforcement Learning, **SL**: Supervised Learning, **I**: Improvement, **BS**: Beam search with width 1280, and **BS***: Beam search with width 1280 and shortest tour heuristic.

As shown in Table 1, our two SL-DABL variants are compared with fourteen representative algorithms. At the top of the table, Concorde (Applegate et al., 2006) and Gurobi (Gurobi Optimization, LLC, 2022) are two exact solvers and LKH3 (Helsgaun, 2017) is a powerful heuristic algorithm. The second group consists of two learn-to-improve algorithms, which are proposed by Wu (Wu et al., 2021) and Costa (d O Costa et al., 2020), respectively. The third group contains two RL-based NCO algorithms, AM (Kool et al., 2018) and POMO (Kwon et al., 2020), each of which has two variants. We also evaluate the performance of POMO trained thought reinforcement learning with data augmentation for clear ablation. The three GCN variants (Joshi et al., 2019) are SL-based two-stage algorithms that generate solutions based on the predicted heatmap. As for Gurobi, LKH3 and the three GCN variants, we directly use the results reported in Joshi et al. (2019) and Fu et al. (2021). We run the other algorithms by ourselves with the codes and pretrained models from their official implementations.

Following the common setting from other NCO work, we separately train three different models for TSP instances with 20-, 50- and 100-node (called TSP20/50/100, respectively). All training datasets are from (Hottung et al., 2020), where each one contains 50,000 TSP instances with optimal solutions solved by Concorde (Applegate et al., 2006). We evaluate the performance and inference time on the test set with 10,000 randomly generated instances for each problem size.

From these results, we can see that SL-DABL outperforms the other nine learning-based algorithms and achieve state-of-the-art results on all three kinds of TSP instances. Especially, SL-DABL × 8 gains the same performance as Concorde on TSP20. Furthermore, our SL-DABL models inherit the real-time inference capability, which is significantly faster than exact and heuristic solvers as well as improvement-based algorithms. It is worth noting that SL-DABL outperforms POMO even though they have the same model structure and inference strategy. These results fully confirm that SL-DABL is a data-efficient and powerful training method for NCO solvers.

## 5.2 ABLATION EXPERIMENT

To investigate the effectiveness of the DA approach and the bidirectional loss, we conduct ablation experiments on each component of SL-DABL. We train the model with four different SL strategies

(i.e., original SL, SL-BL, SL-DA, and SL-DABL) using the dataset of 50,000 labeled TSP instances of size 50. For comparison, we also train the model via RL using 51,000,000 random TSP50 instances. The five trained models are compared on the validation set of 1,000 random instances.

As shown in Figure 1, the original SL is overfitting after about 50 epochs due to the lack of sufficient supervised data. In contrast, the SL-DA optimizes smoothly throughout the training stage without overfitting, and the only difference is that the training dataset is expanded by our proposed DA approach. The bidirectional loss also illustrates its effectiveness, especially when training data is insufficient. In the comparison of original SL and SL-BL, the bidirectional loss helps the latter moderately alleviate overfitting and extract more information from the limited tiny dataset in the early training stage. In the case of the training data containing tremendous labeled data, like SL-DA and SL-DABL, the bidirectional loss can still further improve the data efficiency. SL-DABL is more efficient than SL-DA in getting information from the same amount of training data. SL-DABL and SL-DA have similar performance at the end of training since they are very close to the optimal solutions (e.g., with $0.001\%$ optimal gap). According to the results in Figure 1, the SL-DA and SL-DABL methods have better efficiency than RL throughout the whole training process.

In summary, the original SL approach is inferior to RL mainly due to the overfitting issue with limited training data. Our proposed data augmentation approach can significantly help SL to extract sufficient supervised information from only $50,000$ training instances and thereby address the overfitting issues with negligible cost. Meanwhile, the bidirectional loss can further improve the data efficiency, especially when the training set is relatively small.

## 5.3 GENERALIZATION

The generalization ability is an important concern for the learning-based NCO solver. In real-world applications, the problem instances could typically have different sizes and might come from different distributions. It is crucial that the learning-based solver should still be able to generate good approximate solutions for those unseen instances.

In this section, we compare the generalization ability of SL-DABL and its RL counterpart with the POMO model and different inference strategies. We train two models on TSP100 with SL-DABL and the RL approach in POMO respectively, and then compare their performance on TSP instances with up to 300 nodes. As shown in Table 2, the model trained by SL-DABL always has the better generalization performance.

|  | TSP150 | TSP200 | TSP250 | TSP300 |
|---|---|---|---|---|
| POMO | 9.49834 | 11.00882 | 12.46317 | 13.89798 |
| SL-DABL | 9.43656 | 10.93318 | 12.39314 | 13.84875 |
| POMO × 8 | 9.43073 | 10.91277 | 12.33501 | 13.73801 |
| SL-DABL × 8 | 9.39082 | 10.84699 | 12.26284 | 13.67492 |

Table 2: Generalization of models trained on TSP100

We also test our model on the widely-used TSPlib benchmark (Reinelt, 1991) that contains real-world instances from dramatically different distributions. The statistical results on 30 2-D Euclidean instances with 100 to 300 nodes are shown in Table 3. SL-DABL outperforms its RL counterpart on 23 out of 30 instances and has a better average optimal gap. The detailed results for each instance can be found in Appendix.

| Size | Num | SL-DABL × 8 | | POMO × 8 | |
|---|---|---|---|---|---|
| | | Num | Avg. Gap | Num | Avg. Gap |
| 100-150 | 17 | 13 | 1.462% | 4 | 1.910% |
| 151-200 | 6 | 5 | 4.571% | 1 | 6.723% |
| 201-250 | 3 | 2 | 3.663% | 2 | 4.353% |
| 251-300 | 4 | 3 | 8.204% | 2 | 10.697% |
| 100-300 | 30 | 23 | 3.203% | 7 | 4.286% |

Table 3: Statistical results on TSPlib

## 6 CONCLUSION

In this paper, we have proposed a powerful SL-DABL method for learning traveling salesman problem. It integrates data augmentation to efficiently extract sufficient supervised information from limited training data, and bidirectional loss to better exploit the equivalent properties of optimal solutions. The experiments have validated that SL-DABL can achieve state-of-the-art performance on TSP with only a small set of 50,000 labeled training instances, while also having a better generalization ability to its RL counterpart on real-world instances with various sizes. These findings could be helpful in rethinking the value of efficient SL methods for NCO training.

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

# A  APPENDIX

| Method | Type | TSP20 | | | TSP50 | | | TSP100 | | |
|---|---|---|---|---|---|---|---|---|---|---|
| | | Cost | Gap | Time | Cost | Gap | Time | Cost | Gap | Time |
| Concorde | Solver | 3.82473 | 0.000% | 1.13m | 5.69311 | 0.000% | 6.66m | 7.76541 | 0.000% | 30.85m |
| Gurobi[†] | Solver | 3.8302 | 0.000% | 2.33m | 5.6905 | 0.000% | 26.20m | 7.7609 | 0.000% | 3.57h |
| LKH3[†] | Heuristic | 3.8303 | 0.000% | 20.96m | 5.6905 | 0.001% | 26.25m | 7.7611 | 0.003% | 49.96m |
| Wu {5000} | RL, I | 3.82551 | 0.020% | 1.07h | 5.70456 | 0.201% | 1.42h | 7.88117 | 1.491% | 1.83h |
| Costa {2000} | RL, I | 3.82479 | 0.002% | 28.45m | 5.69957 | 0.114% | 44.23m | 7.82389 | 0.753% | 1.06h |
| AM(Greedy) | RL | 3.83565 | 0.285% | 0.35m | 5.78875 | 1.680% | 0.80m | 8.10442 | 4.366% | 1.80m |
| AM(Sampling) | RL | 3.83694 | 0.319% | 0.42m | 5.80361 | 1.941% | 1.06m | 8.15186 | 4.977% | 2.05m |
| POMO | RL | 3.82557 | 0.022% | 0.01m | 5.70238 | 0.163% | 0.03m | 7.82817 | 0.808% | 0.17m |
| POMO × 8 | RL | 3.82479 | 0.002% | 0.08m | 5.69507 | 0.035% | 0.23m | 7.78218 | 0.216% | 1.08m |
| POMO-DA | RL | 3.82572 | 0.026% | 0.01m | 5.70270 | 0.168% | 0.03m | 7.84934 | 1.081% | 0.17m |
| POMO-DA × 8 | RL | 3.82480 | 0.002% | 0.08m | 5.69521 | 0.037% | 0.23m | 7.78944 | 0.309% | 1.08m |
| GCN[†] | SL | 3.86 | 0.600% | 0.10m | 5.87 | 3.100% | 0.92m | 8.41 | 8.380% | 6m |
| GCN[†] | SL, BS | 3.84 | 0.100% | 0.33m | 5.71 | 0.260% | 2m | 7.92 | 2.110% | 10m |
| GCN[†] | SL, BS[*] | 3.84 | 0.010% | 12m | 5.70 | 0.010% | 18m | 7.87 | 1.390% | 40m |
| SL-DABL(Max) | SL | 3.82502 | 0.008% | 0.01m | 5.69584 | 0.048% | 0.03m | 7.80461 | 0.505% | 0.17m |
| SL-DABL(Max) × 8 | SL | 3.82473 | 0% | 0.08m | 5.69333 | 0.004% | 0.23m | 7.77544 | 0.129% | 1.08m |
| SL-DABL(Mean) | SL | 3.82493 | 0.005% | 0.01m | 5.69525 | 0.038% | 0.03m | 7.7977 | 0.416% | 0.17m |
| SL-DABL(Mean) × 8 | SL | 3.82473 | 0% | 0.08m | 5.69323 | 0.002% | 0.23m | 7.77352 | 0.104% | 1.08m |
| SL-DABL(Min) | SL | 3.82496 | 0.006% | 0.01m | 5.69562 | 0.044% | 0.03m | 7.78793 | 0.290% | 0.17m |
| SL-DABL(Min) × 8 | SL | 3.82473 | 0.000% | 0.08m | 5.69318 | 0.001% | 0.23m | 7.76925 | 0.049% | 1.08m |

Table 4: Experiment results on various TSP instances. In the *Type* column, **RL**: Reinforcement Learning, **SL**: Supervised Learning, **I**: Improvement, **BS**: Beam search with width 1280, and **BS**[*]: Beam search with width 1280 and shortest tour heuristic.

| | TSP150 | TSP200 | TSP250 | TSP300 | TSP400 | TSP500 |
|---|---|---|---|---|---|---|
| RL | 9.49834 | 11.00882 | 12.46317 | 13.89798 | 16.6887 | 19.29866 |
| SL-DABL(Max) | 9.47444 | 10.99099 | 12.44456 | 13.8601 | 16.5457 | 19.0202 |
| SL-DABL(Mean) | 9.46462 | 10.99342 | 12.48445 | 13.96524 | 16.84227 | 19.5274 |
| SL-DABL(Min) | 9.43656 | 10.93318 | 12.39314 | 13.84875 | 16.708 | 19.36179 |
| RL × 8 | 9.43073 | 10.91277 | 12.33501 | 13.73801 | 16.48253 | 19.0658 |
| SL-DABL(Max) × 8 | 9.4142 | 10.89508 | 12.31888 | 13.70788 | 16.36537 | 18.8228 |
| SL-DABL(Mean) × 8 | 9.41023 | 10.89797 | 12.35213 | 13.80193 | 16.64035 | 19.30552 |
| SL-DABL(Min) × 8 | 9.39082 | 10.84699 | 12.26284 | 13.67492 | 16.45711 | 19.06886 |

Table 5: Generalization of models trained on TSP100

## A.1  EXPERIMENTS SETTING

In this work, we plug our SL-DABL into POMO (Kwon et al., 2020), and the details of the model can be found in the corresponding literature. We only describe the hyperparameter settings here, even though all of them are also identical to POMO: there are 100,000 data per epoch and the batch size is 64. The models are optimized by the Adam optimizer in 510 epochs. In the first 500 epochs, the learning rate $\eta = 1e-4$ with a weight decay $w = 1e-6$, while the last 10 epochs fine-tuning the model with $\eta = 1e-5$. The only difference between the training datasets of our SL-DABL and RL is that the former is augmented from a tiny dataset consisting of 50,000 labeled data, while the latter is randomly generated. Both of them use 51,000,000 data to optimize the models. All experiments are implemented on a single Tesla V100 GPU.

## A.2  BIDIRECTIONAL LOSS ABLATION

In this part, we extend the overall comparison with the models trained through the other two aggregated bidirectional loss functions `Max` and `Min`. As shown in Tabel 4, all variants of SL-DABL outperform all other learning-based baselines.

## A.3  ABLATION OF TRAINING DATASET SIZES

| Size | SL-DABL(Min) $\times$ 8 | SL-DABL(Mean) $\times$ 8 | SL-DABL(Max) $\times$ 8 | RL $\times$ 8 |
|---|---|---|---|---|
| kroA100 | 0.475% | 0.296% | 0.700% | 1.001% |
| kroB100 | 0.528% | 0.628% | 0.944% | 0.917% |
| kroC100 | 0.280% | 0.294% | 0.699% | 0.540% |
| kroD100 | 0.329% | 1.437% | 0.498% | 2.268% |
| kroE100 | 0.924% | 1.156% | 0.952% | 0.430% |
| rd100 | 0.607% | 0.670% | 0.607% | 0.721% |
| eil101 | 11.447% | 11.288% | 11.447% | 12.083% |
| lin105 | 0.814% | 1.454% | 1.092% | 1.523% |
| pr107 | 0.553% | 1.205% | 0.851% | 0.973% |
| pr124 | 0.349% | 0.833% | 1.022% | 0.337% |
| bier127 | 1.828% | 1.501% | 5.391% | 3.049% |
| ch130 | 1.277% | 1.538% | 1.489% | 1.358% |
| pr136 | 1.296% | 1.296% | 0.933% | 1.217% |
| pr144 | 0.530% | 1.286% | 1.505% | 0.516% |
| ch150 | 1.762% | 2.436% | 2.252% | 2.068% |
| kroA150 | 1.090% | 1.327% | 1.244% | 1.904% |
| kroB150 | 0.765% | 0.383% | 1.649% | 1.508% |
| pr152 | 1.160% | 2.600% | 1.900% | 1.952% |
| u159 | 0.302% | 0.326% | 0.368% | 1.236% |
| rat195 | 8.653% | 10.891% | 11.580% | 13.173% |
| d198 | 13.219% | 26.857% | 23.169% | 19.195% |
| kroA200 | 2.023% | 3.780% | 2.390% | 1.829% |
| kroB200 | 2.072% | 3.526% | 3.849% | 2.952% |
| ts225 | 1.725% | 4.005% | 2.883% | 4.469% |
| tsp225 | 5.720% | 5.669% | 7.380% | 6.691% |
| pr226 | 3.544% | 7.709% | 4.109% | 1.899% |
| gil262 | 9.504% | 10.050% | 9.546% | 9.420% |
| pr264 | 6.112% | 9.979% | 8.175% | 8.133% |
| a280 | 7.988% | 11.245% | 10.741% | 13.455% |
| pr299 | 9.213% | 12.199% | 13.328% | 11.780% |
| Avg. Gap for 100-150 | 1.462% | 1.708% | 1.957% | 1.907% |
| Avg. Gap for 151-200 | 4.571% | 7.997% | 7.209% | 6.723% |
| Avg. Gap for 201-250 | 3.663% | 5.794% | 4.790% | 4.353% |
| Avg. Gap for 251-300 | 8.204% | 10.868% | 10.448% | 10.697% |
| Avg. Gap for all instances | 3.203% | 4.595% | 4.423% | 4.286% |

Table 6: Results on TSPlib

In this subsection, we study the effect of different numbers of labeled instances (e.g., 10K and 100K) for our proposed SL-DABL method. As shown in Figure 8, the proposed efficient SL-DABL method with only 10k instances will suffer from overfitting, yet which is significantly alleviated compared to the original SL method. In addition, its performance is still better than the reinforcement learning counterpart. In practice, the overfitting issue might be properly handled by early stopping. On the other hand, the improvement on increasing the labeled instances from 50K to 100K is not significant.

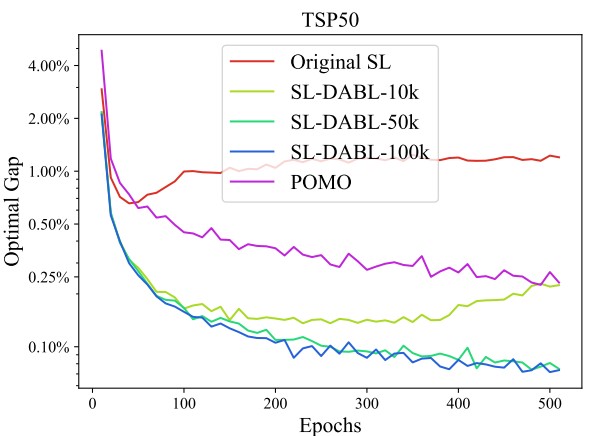

Figure 8: Ablation of training dataset sizes

## A.4 GENERALIZATION

We firstly discuss the generalization of the models trained on TSP100 up to TSP500. As shown in Tabel 5, the SL-DABL with `max` aggregate function superior to RL in all test sets. In addition, the `min` one has better performance on TSP100-TSP300.

The details of the performance on TSBlib are depicted in Tabel 6.

## A.5 CAPACITATED VEHICLE ROUTING PROBLEMS

In this subsection, we investigate the effect of our proposed SL-DABL method on the capacitated vehicle routing problems(CVRP). Most experimental setting is the same with the TSP experiment, and we have $10k$ labeled training instances where the solutions are achieved by the powerful LKH solver (Helsgaun, 2017). We also slightly modified the data augmentation and loss function to accommodate CVRP. For data augmentation, to guarantee the optimality of original solutions, the noise approach is removed since the additional depot hampers the design of the noise upper bound. For the loss function, the sub-tours of given (near-)optimal solutions are reordered to find the most similar one based on the current model. The purpose of this operation is to alleviate the influence of the equivalent solutions. Therefore, different from the $2n$ equivalent solutions in TSP instances, there are $m!2^m$ equivalent solutions for each CVRP instance, where $m$ is the number of sub-tour.

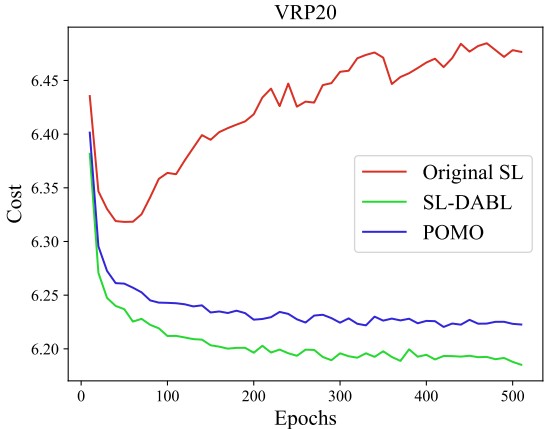

Figure 9: The cost of models trained with different training strategies on the validation set.

As shown in Figure 9, for CVRP20, our SL-DABL significantly mitigates the overfitting compared to the original SL, and the performance is stably superior to POMO. And the numerical result can be checked out in Table 7.

|  | CVRP20 |
|---|---|
| POMO | 6.20833 |
| SL-DABL | 6.18204 |
| POMO $\times$ 8 | 6.16887 |
| SL-DABL $\times$ 8 | 6.15466 |

Table 7: Experiment results on various CVRP instances

### A.6 GCN

In this subsection, we investigate the performance of our proposed data augmentation approaches on the GCN model (Joshi et al., 2019). Since the GCN model directly predicts the solution heatmap as an adjacency matrix, the bidirectional loss functions is not capable in this setting. We follow the same experiment settings as reported in the main paper.As shown in Figure 10, for TSP20, our data augmentation still modestly improve the performance for the GCN model, even if both of them employ the beam search approach with 1280 width.

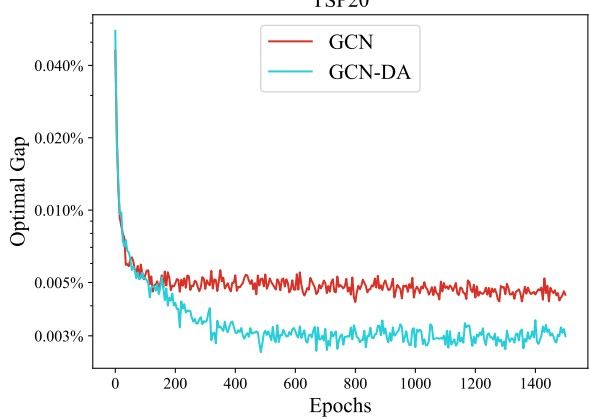

Figure 10: The optimality gap of GCN trained with different training strategies on the validation set.

