# OpenReview forum: "Data-efficient Supervised Learning is Powerful for Neural Combinatorial Optimization"
_ICLR.cc/2023/Conference — Submitted to ICLR 2023_

### Official Review · Reviewer_1fJi · 2022-10-19

**Confidence:** 4
**Correctness:** 4
**Technical Novelty And Significance:** 2
**Empirical Novelty And Significance:** 2
**Recommendation:** 3

**Clarity, Quality, Novelty And Reproducibility:**

Paper is well enough written that the ideas are easy to understand very quickly.

Novelty is the main issue with this work: the two main points of this paper are 1) augmentations help supervised training, 2) but not RL training. The first point is well known, and the paper uses standard rotations, reflections etc. that are all very known symmetries of TSP.

Method seems reproducible enough since hyperparmaeters are explained and the authors implementation builds on an existing implementation for POMO (Kwon et al., 2020). But not code is given and no mention of releasing code is made so it would take significant effort from any reader to attempt to reproduce the results in this paper as the new. components - augmentations etc. - would have to. be implemented from scratch.

**Strength And Weaknesses:**

The underly idea of this work is the following: combinatorial optimization problems often have symmetries, where certain modifications to the problem do not change its solution. Since these modifications do not change the solution they make suitable data augmentations for use during training [or indeed better still the invariances are built into the model architecture]. There may also be modifications of the problem that do change the solution, but change it in a predictable, easily computed way, some of which are considered in this work.

**Strengths**

- Paper is clearly written and easy to quickly ingest the main points and contributions.
- Observation that RL doesn't benefit from augmentations while supervised does is interesting.

**Weaknesses**
- The paper is not suitably contextualized within the contributions of prior work. The appearance is given that augmentations have not been considered before for neural combinatorial optimization. But this is far from true - for instance this paper https://arxiv.org/pdf/2110.10942.pdf also considered data augmentations for TSP. Although I am not certain myself, I would not be surprised if another reviewer is aware of prior work on augmentations and RL for NCO - RL for NCL is a really well studied area so the idea that people do not already know that augmentations are not all that useful sounds unlikely to me [but I will wait on this last point to see if anyone actually knows a suitable reference for this].
- Contributions are stated in a more general form than they actually manifest. Specifically, the authors consistently conflate "neural combinatorial optimization" (NCO) with the TSP problem. For instance the phrase "With these two powerful methods, we propose a novel Supervised Learning with Data Augmentation and Bidirectional Loss (SL-DABL) algorithm for NCO training". This algorithm isn't a general NCO method, it is specific to TSP.
- Evaluation is limited to TSP.


**Summary Of The Paper:**

This paper considers supervised training of neural networks for solving combinatorial optimization problems. They study the effect of data augmentations on generalization. The key finding is that data augmentations are key to successful training (and non0overfitting) of supervised training, whilst reinforcement learning based approaches to learning neural solvers do not benefit from data augmentations.

**Summary Of The Review:**

Whilst the authors are pursuing an important and open question of how to train more effective neural solvers for combinatorial problems, the current paper lacks novelty or conceptual contribution. I would suggest to the authors that they generalize their results to other problems, or seek more depth in their TSP studies.

---

> ### Author Response · Authors · 2022-11-19
> **Response to Reviewer 1fJi [2/2]**
>
> >**3. Data Augmentation for NCO**
>
> Thank you for bringing our attention to this related work [3], which tackles the important issues of adversarial robustness for NCO. In [3], the key finding is that the current NCO solver is sensitive to adversarial examples with small perturbations, and augmenting such adversarial examples in training could improve the solver's robustness for small perturbations (but not for strong perturbations).
>
> In this work, as discussed in the previous response, our main contribution is the novel finding that supervised learning is indeed efficient for training the NCO model, which is against some common beliefs in the NCO community. To improve the training efficiency of supervised learning (especially with limited label solutions), we proposed four data augmentation approaches (and the bidirectional loss design) to fully leverage the invariance/equivariance properties of each training instance. Our contribution is significantly different from [3], and could be novel and surprising to the NCO community.
>
> We do not intend to claim that we are the first to use data augmentation for NCO. We have now added a brief discussion on data augmentation and [3] in Section 2.2. to make this point clear, where data augmentation for efficient NCO inference [4] has already been discussed.
>
> >**4. Reproducibility**
>
> We will release our code and all trained models upon publication.
>
> >**References**
>
> [1] Learning TSP Requires Rethinking Generalization. arXiv preprint arXiv:2006.07054, 2020.
>
> [2] Generalize a Small Pre-trained Model to Arbitrarily Large TSP Instances. AAAI 2021.
>
> [3] Generalization of Neural Combinatorial Solvers Through the Lens of Adversarial Robustness. ICLR 2022.
>
> [4] POMO: Policy Optimization with Multiple Optima for Reinforcement Learning. NeurIPS 2020.

---

> ### Author Response · Authors · 2022-11-19
> **Response to Reviewer 1fJi [1/2]**
>
> Thank you for your constructive and valuable comments and suggestions. We address your concerns point-by-point as follows:
>
> >**1. Novelty and Contribution**
>
> We agree with the reviewer that the data augmentation approach is well-known for supervised learning, and our method was only compared with RL on TSP. However, we want to emphasize that the main contribution of our work is mainly on the new findings on efficient supervised learning for NCO training that against some common beliefs, but not each specific approach. We want to provide a bit more discussion on the motivation and contribution of our approach.
>
> We chose TSP for investigation since it is popular and well-studied in the NCO community, while our findings are indeed surprising and against some current beliefs on NCO for solving TSP. In their well-known work [1], Joshi et al. have shown that supervised learning (even with huge training data) is always outperformed by reinforcement learning in solving TSP for both testing and generalization performance. These results are now well-known and widely recognized in the NCO community. In this work, with simple yet efficient data augmentation and novel bidirectional loss design, we show that supervised learning (only with 4\% data) can indeed outperform reinforcement learning for various problems and settings. We do believe our findings can be valuable and helpful for the community to rethink the role and value of SL and RL for NCO.
>
> The main takeaway we want to provide are: 1) (for TSP) the huge supervised data requirement (a major drawback) is indeed not necessary for supervised learning, and 2) reinforcement learning is not always the best choice for training an NCO model. In this work, we mainly conduct a set of carefully designed experiments on TSP, but we believe these findings and the general principle of invariance/equivariance properties could still be valuable for other combinatorial optimization problems. We notice that some NCO works, such as [1,2], only focus on TSP in their paper, but their findings could be beneficial for more general NCO methods. We sincerely hope our findings can inspire more work on efficient supervised learning for different NCO problems.
>
> Following the reviewer's comments, we will:
>
> + Revise the paper accordingly to make the statement precise and accurate;
>
> + We plan to change our title to: Data-efficient Supervised Learning is Powerful for Learning Traveling Salesman Problem. Any suggestions are welcome.
>
> + Report results on CVRP and new model (see below).
>
> >**2. More Problems**
>
> To show our proposed methods can be generalized to different problem/settings, we run new experiments on 1) the POMO framework for CVRP and 2) the GCN framework for TSP, both with limited training data. The whole sets of experiments need more time to finish. We now report the results on CVRP20/TSP20 in Appendix A.5/A.6 and will add the results for CVRP50/CVRP100 and TSP50/TSP100 once the experiments are done. The current results still consistently support our claim that data-efficient supervised learning is promising for NCO.

---

### Official Review · Reviewer_5UpA · 2022-10-23

**Confidence:** 3
**Correctness:** 4
**Technical Novelty And Significance:** 3
**Empirical Novelty And Significance:** 3
**Recommendation:** 6

**Clarity, Quality, Novelty And Reproducibility:**

The paper is clear and of high quality. The ideas are novel in the ML for CO literature, but are borrowed from concepts that are increasingly prevalent in other deep learning settings.

**Strength And Weaknesses:**

Strengths
- The overall idea is intuitive and simple. The augmentations are easy to implement and the bidirectional loss is neat.

Weaknesses
- In Algorithm 5, what is the motivation for the augmentation procedure of rotation or symmetry + shrink + noise? An intuitive augmentation procedure would be to employ random chance of rotation + random chance of symmetry + random chance of shrink + random chance of noise.
- It seems like the strategies are geared towards TSP problems, but what about other combinatorial optimization problems? Do the frameworks scale to all CO problems that can be formulated on graphs?

**Summary Of The Paper:**

Neural Combinatorial Optimization (NCO) with supervised learning suffers from data inefficiency when applied to the Traveling Salesman Problem (TSP). This paper proposes a series of data augmentations using properties such as rotation and symmetry invariance of TSP solutions in 2-D, along with a bidirectional loss function. With only 50,000 training examples, the proposed algorithm achieves SOTA on 2-D TSP.


**Summary Of The Review:**

Overall, the paper is interesting and presents intuitive methods. It is easy to read and the results are well-supported. My key question remains on whether it is generalizable beyond TSPs only.

---

> ### Author Response · Authors · 2022-11-19
> **Response to Reviewer 5UpA**
>
> Thank you for your constructive and valuable comments and suggestions. We address your concerns point-by-point as follows:
>
> >**1. Novelty**
>
> We agree with the reviewer that our proposed methods (e.g., data augmentation) are borrowed from concepts that are increasingly prevalent in other deep learning settings, and it was only compared with RL on TSP. However, we want to emphasize that the main contribution of our work is mainly on the new findings on supervised learning for NCO training, but not each specific approach. We want to provide a bit more discussion on the motivation and contribution of our approach.
>
> We choose TSP as the studied problem since it is popular and well-studied in the NCO community, while our findings are indeed surprising and against some current beliefs on NCO for solving TSP. In their well-known work [1], Joshi et al. have shown that supervised learning (even with huge training data) is always outperformed by reinforcement learning on solving TSP for both testing and generalization performance. These results are now well-known and widely recognized in the NCO community. In this work, with simple yet efficient data augmentation and novel bidirectional loss design, we show that supervised learning (only with 4\% data) can indeed outperform reinforcement learning for various problems and settings. We believe our findings could be valuable and helpful for the community to rethink the role and value of SL and RL for NCO.
>
> The main takeaway we want to provide are: 1) (for TSP) the huge supervised data requirement (a major drawback) is indeed not necessary for supervised learning, and 2) reinforcement learning is not always the best choice for training an NCO model. In this work, we mainly conduct a set of carefully designed experiments on TSP, but we believe these findings and the general principle of invariance/equivariance properties could still be valuable for other combinatorial optimization problems. We notice that some NCO works, such as [1,2], only focus on TSP in their paper, but their findings could be beneficial for more general NCO methods. We sincerely hope our findings can inspire more work on efficient supervised learning for different NCO problems.
>
> >**2. Generalization to Other Problems**
>
> To show our proposed methods can be generalized to different problem/settings, we run new experiments on 1) the POMO framework for CVRP and 2) the GCN framework for TSP, both with limited training data. The whole sets of experiments need more time to finish. We now report the results on CVRP20/TSP20 in Appendix A.5/A.6 and will add the results for CVRP50/CVRP100 and TSP50/TSP100 once the experiments are done. The current results still consistently support our claim that efficient supervised learning with data augmentation is promising for NCO.
>
> >**3. The motivation for the augmentation procedure of rotation or symmetry + shrink + noise**
>
> The symmetry operation also includes random rotation. Through the symmetry operation, an instance is flipped with a certain probability after being randomly rotated. We set the flipping probability to 0.5 to obtain a dataset with more diversity, rather than a dataset that is either fully flipped or non-flipped. That is, our augmentation procedure includes random rotation + 50/50 chance flipping + shrink + noise.
>
>
> >**4. Reproducibility**
>
> We will release our code and all trained models upon publication.
>
> >**References**
>
> [1] Learning TSP Requires Rethinking Generalization. arXiv preprint arXiv:2006.07054, 2020.
>
> [2] Generalize a Small Pre-trained Model to Arbitrarily Large TSP Instances. AAAI 2021.

---

### Official Review · Reviewer_VFpH · 2022-10-24

**Confidence:** 4
**Correctness:** 3
**Technical Novelty And Significance:** 2
**Empirical Novelty And Significance:** 2
**Recommendation:** 5

**Clarity, Quality, Novelty And Reproducibility:**

The paper writing is clear, and the approach is sound. Although the code is not provided, the results should be reproducible given the simplicity of their approach.

However, in general I think the novelty of this work is limited. Data augmentation is not a new idea for general machine learning community. The proposed SL-DABL algorithm is specific to TSP, and I don't think it is extensible to general combinatorial optimization problems. Also, this work is not the first one considering equivariance properties of TSP, and there are prior works leveraging these properties at the inference time [1] and in the architecture design [2]. Thus, the technical significance is not enough.

[1] Kwon et al., POMO: Policy Optimization with Multiple Optima for Reinforcement Learning, NeurIPS 2020.
[2] Ouyang et al., Generalization in Deep RL for TSP Problems via Equivariance and Local Search.

**Strength And Weaknesses:**

Strengths:

1. The proposed approach is simple yet effective for solving TSP with neural networks, and it is model-agnostic.

2. Their approach achieves better generalization to larger TSP instances compared to the POMO baseline.

Weaknesses:

Overall, I think the proposed approach is specific to TSP. Both data augmentation and bidirectional loss design leverage the properties of TSP itself, and are not applicable to most other combinatorial optimization problems. To show that the approach is effective for more problems, the authors should also present results on Capacitated Vehicle Routing problems (CVRP), where there have been several works on designing learning algorithms for CVRP (e.g., [1][2][3][4]).

Meanwhile, I wonder whether the proposed SL-DABL algorithm is effective for different training data sizes. In general, the advantage of RL approaches is that they do not require optimal solutions for training. While 50K samples in the current setting is smaller than some prior works, it is not a small number either. Therefore, it is helpful to study how SL-DABL compares to other baselines when the training size is even smaller, and whether the approach can achieve better performance with more training data.

Also, the authors should study the effect of SL-DABL with different model architectures, especially those with some equivariance properties already incorporated in the architectural design. For example, the authors can evaluate on GCN and see whether their training algorithm can improve the performance.

[1] Kool et al., Attention, Learn to Solve Routing Problems! ICLR 2019.
[2] Nazari et al.,  Reinforcement Learning for Solving the Vehicle Routing Problem, NeurIPS 2018.
[3] Chen and Tian, Learning to Perform Local Rewriting for Combinatorial Optimization, NeurIPS 2019.
[4] Kwon et al., POMO: Policy Optimization with Multiple Optima for Reinforcement Learning, NeurIPS 2020.

**Summary Of The Paper:**

This paper proposes a supervised learning algorithm called Supervised Learning with Data Augmentation and Bidirectional Loss (SL-DABL) for traveling salesman problems (TSP). There are two components in the approach. The data augmentation method leverages several equivalence properties of traveling TSP, e.g., the optimal solutions stay the same for problems transformed with rotation, horizontal flip, scaling, etc. The bidirectional loss leverages the fact that the same route can be represented in different ways, e.g., via starting from different nodes. They evaluate their approach in the setting where there are 50K training samples with optimal solutions, and compare their approach to several reinforcement learning and supervised learning baselines as well as the non-learning-based solvers. They demonstrate that their approach achieves better optimality gap compared to learning-based approaches, while the inference time is also much shorter than classic non-learning solvers for combinatorial optimization problems.

**Summary Of The Review:**

This work presents good empirical results for TSP compared to prior learning-based approaches. However, the proposed approach is specific to TSP, and is not generally applicable to other combinatorial optimization problems. Meanwhile, the evaluation setting is a bit restricted, and some aspects of the approach are not well-studied. Therefore, I recommend a rejection.


----------
I thank the authors for the response and new experiments. I understand that the time for revision and adding experiments is short, but I still feel that the current experiments are insufficient and the scope of this work is limited. So I keep my score.

---

> ### Author Response · Authors · 2022-11-19
> **Response to Reviewer VFpH**
>
> Thank you for your constructive and valuable comments and suggestions. We address your concerns point-by-point as follows:
>
> >**1. Contribution and Novelty**
>
> We agree with the reviewer that the data augmentation is not a new idea for the general machine learning community, and our proposed SL-DABL algorithm was only compared with RL on TSP. However, we want to emphasize that the main contribution of our work is mainly on the new findings on supervised learning for NCO training, but not each specific approach. We want to provide a bit more discussion on the motivation and contribution of our approach.
>
> We chose TSP for investigation since it is popular and well-studied in the NCO community, while our findings are indeed surprising and against some current beliefs on NCO for solving TSP. In their well-known work [1], Joshi et al. have shown that supervised learning (even with huge training data) is always outperformed by reinforcement learning in solving TSP for both testing and generalization performance. These results are now well-known and widely recognized in the NCO community. In this work, with simple yet efficient data augmentation and novel bidirectional loss design, we show that supervised learning (only with 4\% data) can indeed outperform reinforcement learning for various problems and settings. We do believe our findings can be valuable and helpful for the community to rethink the role and value of SL and RL for NCO.
>
> The main takeaway we want to provide are: 1) (for TSP) the huge supervised data requirement (a major drawback) is indeed not necessary for supervised learning, and 2) reinforcement learning is not always the best choice for training an NCO model. In this work, we mainly conduct a set of carefully designed experiments on TSP, but we believe these findings and the general principle of invariance/equivariance properties could still be valuable for other combinatorial optimization problems. We notice that some NCO works, such as [1,2], only focus on TSP in their paper, but their findings could be beneficial for more general NCO methods. We sincerely hope our findings can inspire more work on efficient supervised learning for different NCO problems.
>
> >**2. Different Problem and Model Architectures**
>
> To show our proposed methods can be generalized to different problem/setting, we run new experiments on 1) the POMO framework for CVRP and 2) the GCN framework for TSP, both with limited training data. The whole sets of experiments need more time to finish. We now report the results on CVRP20/TSP20 in Appendix A.5/A.6 and will add the results for CVRP50/CVRP100 and TSP50/TSP100 once the experiments are done. The current results still consistently support our claim that efficient supervised learning with data augmentation is promising for NCO.
>
> >**3. Different Training Data Sizes**
>
> Following your suggestion, we have conducted experiments with different numbers of labeled instances (e.g., 10k and 100k) in Appendix A.3. According to the results, we find that the proposed efficient SL-DABL method with only 10k instances will suffer from overfitting, yet which is significantly alleviated compared to the original SL method. In addition, its performance is still better than the reinforcement learning counterpart. In practice, the overfitting issue might be properly handled by early stopping. On the other hand, the improvement on increasing the labeled instances from 50K to 100K is not significant.
>
> >**4. Reproducibility**
>
> We will release our code and all trained models upon publication.
>
> >**References**
>
> [1] Learning TSP Requires Rethinking Generalization. arXiv preprint arXiv:2006.07054, 2020.
>
> [2] Generalize a Small Pre-trained Model to Arbitrarily Large TSP Instances. AAAI 2021.

---

### Official Review · Reviewer_3o4f · 2022-10-24

**Confidence:** 4
**Correctness:** 4
**Technical Novelty And Significance:** 2
**Empirical Novelty And Significance:** 2
**Recommendation:** 5

**Clarity, Quality, Novelty And Reproducibility:**

The paper is clearly written and technically correct.

However, the data augmentation method and bidirectional loss proposed in the paper are not very novel in my opinion. Please see the weaknesses above.

Since the paper is mainly based on the open-source POMO, I think the reproducibility of the paper should be ok.

**Strength And Weaknesses:**

Strength

1. The paper proposes to train the supervised learning (SL) method with the POMO framework, i.e., autoregressive decoding, and show that with data augmentation and a novel loss objective, SL can actually achieve better performance than reinforcement learning (RL), which overthrows the conclusion from [1].

2. The empirical results also show that the SL method trained on TSP-100 can achieve better performance on TSP-150/200/250/300 than RL (i.e., POMO) method, which again overthrows the conclusion from [1].

3. Overall, I believe the results from this paper could help the NCO community rethink the role and value of SL and RL.


Weaknesses

1. My main concern is the empirical effectiveness of data augmentation. As shown in the paper, in four types of data augmentation in training (i.e., Rotation, Symmetry, Shrink, Noise), rotation and symmetry are the most effective ones. However, in the graph neural network community, equivariant neural networks [2,3] can directly model an equivariant function without any data augmentation and significantly outperforms data augmentation methods. I am wondering if the authors have considered these equivariant neural networks.

2. The idea of minimizing the minimal loss from a set of candidates is not new in the literature of combinatorial optimization. For example, [4] applied it to the minimum independent set problem.


Typos:

Page 3: "Kwon et al. (2020) has discussed how to use the data augmentation methods to improve the inference
performance for NCO but not for RL-based training." ==> "have", "but not for *SL*-based training."


[1] Chaitanya K Joshi, Quentin Cappart, Louis-Martin Rousseau, and Thomas Laurent. Learning tsp requires rethinking generalization. arXiv preprint arXiv:2006.07054, 2020.

[2] N. Thomas, T. Smidt, Steven M. Kearnes, Lusann Yang, L. Li, Kai Kohlhoff, and P. Riley. Tensor field networks: Rotation- and translation-equivariant neural networks for 3d point clouds. ArXiv, 2018.

[3] Victor Garcia Satorras, Emiel Hoogeboom, and Max Welling. E(n) equivariant graph neural networks, 2021b.

[4] Li, Zhuwen, Qifeng Chen, and Vladlen Koltun. "Combinatorial optimization with graph convolutional networks and guided tree search." Advances in neural information processing systems 31 (2018).

**Summary Of The Paper:**

The paper proposes a new supervised-learning method for combinatorial optimization problems. One of the main contributions of the paper is the introduction of four types of data augmentation in training (i.e., Rotation, Symmetry, Shrink, and Noise). Another contribution is a new bidirectional loss, in which given the optimal solution, the training process will only minimize the autoregressive likelihood for the starting point and direction of minimal loss. Experiments show that due to more informative gradients from the annotation, the supervised neural solver can outperform reinforcement learning methods on TSP-20/50/100 problems.

**Summary Of The Review:**

The paper provides a strong empirical study of data-efficient supervised learning for NCO, and could help the NCO community rethink the role and value (e.g., generalization & convergence speed) of SL and RL. However, the data augmentation method and bidirectional loss proposed in the paper are not very novel in my opinion.

Therefore, I would recommend a score of 5 for the paper.

---

> ### Author Response · Authors · 2022-11-19
> **Response to Reviewer 3o4f**
>
> Thank you for your constructive and valuable comments and suggestions. We address your concerns point-by-point as follows:
>
> > **1. Empirical Effectiveness of Data Augmentation**
>
> Thank you for bringing our attention to the equivariant neural networks. We believe they have great potential and could be strong models for NCO. In this work, instead of building more powerful models, we focus on improving the supervised learning approach, and study whether it can outperform the reinforcement learning counterpart on **the same model** (e.g., Attention Model structure with the POMO approach). We want to provide a bit more discussion on the motivation and contribution of our approach.
>
> In the current NCO work, there is a common belief that reinforcement learning could be better than supervised learning for training the same NCO model. In their well-known work [1], Joshi et al. have shown that supervised learning (even with huge training data) is always outperformed by reinforcement learning on solving TSP for both testing and generalization performance. These results are now widely recognized in the NCO community. In this work, with simple yet efficient data augmentation and novel bidirectional loss design, we show that supervised learning (only with 4\% data) can indeed outperform reinforcement learning for various problems and settings. Our findings could be valuable and helpful for the community to rethink the role and value of SL and RL for NCO.
>
> We believe these main messages of our work (e.g., efficient SL over RL) could be best delivered with experiments on an identical and well-known NCO model. Therefore, we left the study of equivariant neural networks for NCO to future work (with potential SL v.s. RL comparison). A brief discussion on the equivariant neural networks has now been added in section 2.2.
>
> >**2. Novelty**
>
> We agree with the reviewer that the data augmentation method and bidirectional loss design themselves are both simple and not very novel.
>
> However, we want to emphasize that the main contribution of our work is mainly on the new findings on supervised learning for NCO training, but not each specific approach. The main takeaway we want to provide are: 1) (for TSP) the huge supervised data requirement (a major drawback) is indeed not necessary for supervised learning, and 2) reinforcement learning is not always the best choice for training an NCO model. Our findings, which against some common belief, could be surprising and valuable for many researchers working on NCO, and hence we believe it is a novel contribution to the community.
>
> >**3. Reproducibility**
>
> We will release our code and all trained models upon publication.
>
> >**4. Typos**
>
> Thank you for pointing them out, we have fixed all of them in the revised paper.
>
> >**References**
>
> [1] Learning TSP Requires Rethinking Generalization. arXiv preprint arXiv:2006.07054, 2020.

---

### Official Review · Reviewer_Ke1k · 2022-11-01

**Confidence:** 4
**Correctness:** 4
**Technical Novelty And Significance:** 2
**Empirical Novelty And Significance:** 2
**Recommendation:** 5

**Clarity, Quality, Novelty And Reproducibility:**

- The paper is very well written and the proposed ideas and claims are clear.
- Both the theory and the experiments are generally well supported and the paper does not make unsubstantiated claims. One exception regards the bold statement about NCO. The scope of the paper is limited, and I do not personally see how it shows that NCO in general can benefit from supervised learning.
- The novelty of the work is not strong. The ideas are sound but based on rather simple invariances/symmetries of the TSP problem. That said, they seem to result in much improved optimality gaps.
- The authors have run the other algorithms by themselves with the codes and pretrained models from their official implementations. Regarding their framework, they have not released the code yet, but their results could be reproduced if they later do so, since the experimental details are provided in the appendix.

**Strength And Weaknesses:**

Strength
- The paper is well written and simple to follow. The various ideas are clearly stated and written down in sufficient detail.
- The motivation is interesting. RL may not necessarily be the best choice for training, as the authors argue. They provide an interesting alternative and are able to outperform POMO training with RL. Their results on TSP20, TSP50 and TSP100 are state-of-the-art compared to the surveyed approaches and thus supportive of the proposed supervised framework.
- Despite being simple, the proposed ideas are based on sound theoretical fundamentals (e.g., invariance of optimal solutions under certain transformations).
- The experimental compares to various solvers (traditional and NCO). Furthermore, the ablation study sheds light on the individual components of the proposed framework.

Weaknesses
- The paper makes a bold statement already in the title, i.e., that supervised learning is powerful for NCO. However, both the theory and the experiments are specifically tailored for the TSP problem. I am concerned that this big claim is not really supported in the paper. The scope of the paper is limited to TSP, and any evidence in favor of NCO will necessarily come from TSP only. This is not just about being precise. My main source of concern is that the proposed data augmentations and bidirectional loss are mainly meaningful for the TSP, but may not be applicable for other challenging combinatorial problems such as capacitated vehicle routing problem, or the 0-1 knapsack problem. If the authors wanted to make such a general statement, then they should have studied more problems (with possibly different sets of data augmentations and invariances/symmetries), and show that supervised learning is powerful for these problems, too.
- Related to the problem above, the scope is limited. I think the paper would have benefitted from including more problems, or alternatively, by arguing how the proposed ideas are more widely applicable to problems other than TSP. In the current exposition, it seems to me that the story is almost exclusively about TSP.
- The generalization results are somehow encouraging - indeed, compared to POMO with RL, the proposed framework achieves better results. However, I am not convinced that the proposed symmetries can inherently improve generalization to instances of varying lengths. I think they should enhance learning efficiency for instances of a fixed length, and this is aligned with the results on TSP20, TSP50, and TSP100. But I do not really see how their design can inherently lead to improved generalization. Also, note that POMO with RL suffers from poor generalization power - so, even if the proposed framework improves upon POMO with RL, this alone is not necessarily a sign of good generalization power.
- The novelty is not strong. The proposed framework is mostly based on simple properties of the TSP problem. This is not necessarily a problem by itself, but it could be an issue when coupled with the limited scope of the paper.

**Summary Of The Paper:**

The paper develops a methodology based on supervised learning (as opposed to reinforcement learning) for the purpose of neural combinatorial optimization (NCO), and in particular for the traveling salesman problem (TSP). In this direction, the authors first propose a set of four data augmentation methods (rotation, symmetry, shrink and noise), which augments the original set of data without affecting the label of the optimal solution. This allows to extract sufficient information from a small set of high-quality labeled solutions.  Subsequently, the authors introduce a new bidirectional supervised loss, which  leverages the equivalence of solutions for the TSP, e.g., a path remains an optimal solution is we reverse the directionality of all edges, or if we cyclically shift the path by any amount. The paper then builds upon these two ideas and introduces the SL-DABL algorithm for NCO training.

The authors then conduct an extensive experimental study, where they train POMO with their supervised learning framework (instead of reinforcement learning), and show that they can achieve state-of-the-art results with only 50,000 high-quality training instances. Furthermore, they provide encouraging results of better generalization performance to test instances of different sizes that the training instances.

**Summary Of The Review:**

The paper is well-written and proposes a sound framework for improving the TSP problem with supervised learning. The empirical results look good, even though I am not personally clear that the generalization power for the new framework is strong. The main reason why I am on the fence is because I feel that both the scope and the novelty are not significant, which is a concern for a top-tier venue like ICLR.

---

> ### Author Response · Authors · 2022-11-19
> **Response to Reviewer Ke1k [2/2]**
>
> > **4. Novelty**
>
> We agree with the reviewer that our proposed data augmentation approach and bidirectional supervised loss design are both simple and straightforward. We are also glad to know the reviewer also believe simplicity itself is not a weakness.
>
> As discussed in the previous response, the goal of our work is to show that a data-efficient supervised learning approach is indeed powerful to train NCO model for solving TSP. Our findings which against some common belief could be surprising and valuable for many researchers working on NCO, and hence we believe it is a novel and significant contribution to the community.
>
> > **5. Reproducibility**
>
> We will release our code and all trained models upon publication.
>
> > **References**
>
> [1] Learning TSP Requires Rethinking Generalization. arXiv preprint arXiv:2006.07054, 2020.
>
> [2] Generalize a Small Pre-trained Model to Arbitrarily Large TSP Instances. AAAI 2021.

---

> > ### Comment · Reviewer_Ke1k · 2022-12-13
> > **paper can benefit from a major revision**
> >
> > I thank the authors for their detailed rebuttal. Overall, I believe the framework is interesting and promising. That said, there are issues related to the paper's scope. The current focus is predominantly on TSP - that might be OK if the paper was particularly novel and offered brand new insights but I do not think tis is the case. The authors added some experiments in the appendix, but a more through investigation would be needed for definitive conclusions. I believe the paper would benefit from a new revision, where these issues would be addressed in depth.

---

> ### Author Response · Authors · 2022-11-19
> **Response to Reviewer Ke1k [1/2]**
>
> Thank you for your constructive and valuable comments and suggestions. We address your concerns point-by-point as follows:
>
>
> >**1. Scope and Contributions**
>
> Thank you for pointing this out. We agree with the reviewer that our work currently focuses on NCO for TSP but not for a general combinatorial optimization problem. Before introducing the concrete revision we will make, we want to briefly discuss this work's scope and contribution.
>
> We chose TSP for investigation since it is popular and well-studied in the NCO community, while our findings are indeed surprising and against some current beliefs on NCO for solving TSP. In their well-known work [1], Joshi et al. have shown that supervised learning (even with huge training data) is always outperformed by reinforcement learning in solving TSP for both testing and generalization performance. These results are now well-known and widely recognized in the NCO community. In this work, with simple yet efficient data augmentation and novel bidirectional loss design, we show that supervised learning (only with 4\% data) can indeed outperform reinforcement learning for various problems and settings. We do believe our findings can be valuable and helpful for the community to rethink the role and value of SL and RL for NCO.
>
> The main takeaway we want to provide are: 1) (for TSP) the huge supervised data requirement (a major drawback) is indeed not necessary for supervised learning, and 2) reinforcement learning is not always the best choice for training an NCO model. In this work, we mainly conduct a set of carefully designed experiments on TSP, but we believe these findings and the general principle of invariance/equivariance properties could still be valuable for other combinatorial optimization problems. We notice that some NCO works, such as [1,2], only focus on TSP in their paper, but their findings could be beneficial for more general NCO methods. We sincerely hope our findings can inspire more work on efficient supervised learning for different NCO problems.
>
> Following the reviewer's comments, we will:
>
> + Revise the paper accordingly to make the statement precise and accurate;
>
> + We plan to change our title to: Data-efficient Supervised Learning is Powerful for Learning Traveling Salesman Problem. Any suggestions are welcome.
>
> + Report results on CVRP and new model (see below).
>
>
> >**2. More Problems**
>
> To show our proposed methods can be generalized to different problem/settings, we run new experiments on 1) the POMO framework for CVRP and 2) the GCN framework for TSP, both with limited training data. The whole sets of experiments need more time to finish. We now report the results on CVRP20/TSP20 in Appendix A.5/A.6 and will add the results for CVRP50/CVRP100 and TSP50/TSP100 once the experiments are done. The current results still consistently support our claim that data-efficient supervised learning is promising for NCO.
>
>
> >**3. Generalization**
>
> Thank you for pointing this out. The goal of the generalization experiments is to investigate a common belief that supervised learning can always be outperformed by reinforcement learning for generalization performance. Our results show that, with our proposed methods, the efficient supervised learning approach (even with only 50k labeled data) can indeed achieve promising performances on unseen tasks with different sizes or different distributions from the real-world TSPlib problems). Therefore, compared with the reinforcement learning counterpart, supervised learning will not hurt the generalization performance.
>
> However, it could be much more challenging to design a theoretical solid approach to enhance the generalization ability for NCO (on both supervised learning and reinforcement learning). We agree with the reviewer that there is no clear connection from the symmetry property to the better generalization performance on the problem with varying lengths, and the POMO with RL approach actually suffers from poor generalization performance. We will revise the claim on generalization performance carefully, and make our statement clear.

---

### Decision · Program_Chairs · 2023-01-20

**Decision:**

Reject

**Justification For Why Not Higher Score:**

Reviewers are in general consensus that the paper in its current form is not ready for publication. AC finds no evidence to override this decision.

**Justification For Why Not Lower Score:**

N/A

**Metareview: Summary, Strengths And Weaknesses:**

This paper studies a neural combinatorial solver for the TSP problem. The main contributions lie in the diverse set of data augmentations that facilitate more data-efficient neural solver. Reviewers are in general consensus that the paper is not ready for acceptance. While the paper has its merits in studying meaningful techniques specialized to the TSP problem, the flaws are outweighing the merits. Reasons are the narrow scope of the solver to TSP problems, the narrow applicability of the augmentation tricks to other CO problems. In conclusion, the scope is so narrow that a more pronounced technical method with enhanced novelty is expected, or at least more sufficient empirical study is needed. However, we can see neither. The preliminary study on CVRP problems of very small size cannot lead to definitive conclusion that the approach is of high practical utility. Hence, the paper needs a major revision to be considered publishable at a top venue.